# Multifunctional Monoclonal Antibody Targeting *Pseudomonas aeruginosa* Keratitis in Mice 

**DOI:** 10.3390/vaccines8040638

**Published:** 2020-11-02

**Authors:** Wesley Hebert, Antonio DiGiandomenico, Michael Zegans

**Affiliations:** 1Department of Microbiology and Immunology, Geisel School of Medicine at Dartmouth School, Hanover, NH 03756, USA; wesley.p.hebert@hitchcock.org; 2Microbial Sciences, BioPharmaceuticals R&D, AstraZeneca, One MedImmune Way, Gaithersburg, MD 20878, USA; antonio.digiandomenico@astrazeneca.com

**Keywords:** bacterial keratitis, microbial keratitis, antimicrobial, immunotherapy, pseudomonas aeruginosa, cornea, antibody

## Abstract

A worrisome trend in the study and treatment of infectious disease noted in recent years is the increase in multidrug resistant strains of bacteria concurrent with a scarcity of new antimicrobial agents to counteract this rise. This is particularly true amongst bacteria within the *Enterococcus faecium*, *Staphylococcus aureus*, *Klebsiella pneumoniae*, *Acinetobacter baumannii*, *Pseudomonas aeruginosa*, and Enterobacter species (ESKAPE) designation. *P. aeruginosa* is one of the most common causes of bacterial keratitis. Therefore, it is of vital importance to characterize new antimicrobial agents with anti-Pseudomonal activity for use with the ocular surface. MEDI3902 is a multifunctional antibody that targets the *P. aeruginosa* persistence factor Psl exopolysaccharide, and the type 3 secretion protein PcrV. We initially assessed this antibody for ocular surface toxicity. The antimicrobial activity of the antibody was then tested by treating mice with established *P. aeruginosa* keratitis with both topical and intravenous treatment modalities. MEDI3902, was shown to be non-toxic to the ocular surface of mice when given topically. It was also effective compared to the control antibody at preventing *P. aeruginosa* keratitis with a one-time treatment at the time of infection. Both topical and intravenous administration of MEDI3902 has been proved significant in treating established keratitis infections as well, speeding the resolution of infection significantly more than that of the control IgG. We report the first use of a topical immunotherapeutic multifunctional agent targeting Psl and type 3 secretion on the ocular surface as an antimicrobial agent. While MEDI3902 has been shown to prevent *Pseudomonas* biofilm formation in keratitis models when given prophylactically intravitally, we show that MEDI3902 has the capability to also treat an active infection when given intravenously to mice with *Pseudomonas* keratitis. Our data indicate antibodies are well tolerated and nontoxic on the ocular surface. They reduce infection in mice treated concurrently at inoculation and reduced the signs of cornea pathology in mice with established infection. Taken together, these data indicate treatment with monoclonal antibodies directed against Psl and PcrV may be clinically effective in the treatment of *P. aeruginosa* keratitis and suggest that the design of further antibodies to be an additional tool in the treatment of bacterial keratitis.

## 1. Introduction

There is a need for novel broad-spectrum antimicrobials in a variety of clinical settings [1], including for the treatment of ocular surface infection [2,3]. This need stems from the limited number of antimicrobials approved for use on the ocular surface. This scarcity is due in part because of the ocular incompatibility of certain antibiotics commonly administered orally or systemically. Penicillins are seldom used as eye drops and are not commercially available because of a history of severe allergic reactions when given topically. Furthermore, there is less economic incentive for developing antimicrobials for very specific indications such as delivery to the ocular surface. At the same time, broad-spectrum ocular antimicrobials, which were once effective against most ocular bacterial pathogens, are now not as effective because of the development of drug resistance [4]. When the second-generation fluoroquinolone ciprofloxacin was introduced in the early 1990s, it had wide-ranging activity against Gram-positive and Gram-negative pathogens. However, since its introduction, resistance to ciprofloxacin particularly among Gram-positive bacteria has increased among ocular isolates to the point where it is no longer clinically reliable [5].

Bacterial keratitis is common. An analysis in 2010 by the Centers for Disease Control and Prevention identified 930,000 office visits and 58,000 emergency room visits related to keratitis. Seventy-five percent of these visits resulted in antimicrobial prescriptions resulting in 175 million dollars in direct health care payments [6]. Bacterial keratitis is 10 to 70 times more common in developing countries and can be a significant cause of monocular blindness [7,8,9]. *Pseudomonas* spp., *Staphylococcus* spp., and *Streptococcus* spp. are the predominant pathogens associated with bacterial keratitis. *P. aeruginosa* is consistently one of the top 3 pathogens recovered from corneal ulcers. Among contact lens related ulcers, *P. aeruginosa* is often the most common pathogen and accounts for many of the most severe, vision-threatening cases [10,11].

Bacteria can produce exopolysaccharides (EPSs) which are secreted sugars that play a complex role in bacterial physiology and pathogenesis. *P. aeruginosa* is capable of producing at least 3 EPSs; Psl, Pel, and alginate [12]. Regulation of EPS production appears to participate in the pathogenesis of specific infections [13]. In the Steroids for Corneal Ulcers Trial (SCUT), 99% of *P. aeruginosa* keratitis isolates produced Psl EPS and 100% of the evaluated Psl positive isolates were susceptible to anti-Psl mAb-mediated opsonophagocytic killing (OPK) [14]. This suggests that anti-Psl monoclonal antibodies might be effective in preventing or treating bacterial keratitis. Previously published studies indicated systemic pretreatment with anti-Psl monoclonal antibodies prevented the development of keratitis in a murine model [15].

Bacterial species such as *Pseudomonas*, *Shigella*, *Yersinia*, *Salmonella*, and other Gram-negatives contain a Type III Secretion System (T3SS). This injectosome acts as a “molecular syringe” to inject host cells with cytotoxins [16]. In *P. aeruginosa*, the T3SS consists of a secretion apparatus and four virulence factors known as exoenzymes: ExoS, ExoT, ExoU, and ExoY [17,18,19]. The PcrV protein is thought to reside at the tip of the secretion apparatus. The T3SS has been associated with greater risk of death in systemic and lower respiratory *P. aeruginosa* infections. Similar to anti-Psl antibodies, it was demonstrated that an anti-PcrV monoclonal antibody protects from lower respiratory infections and effectively treats sepsis caused by this pathogen in preclinical disease models and enhances survivability in mice with burns [20,21,22]. However, the efficacy and/or use of an anti-PcrV antibody in the ocular setting remains untested; although topical treatment of fungal keratitis with antibodies has been demonstrated in mice [23].

Both anti-Psl and anti-PcrV specificities were combined in the bispecific mAb MEDI3902. This antibody was shown to maintain anti-Psl and anti-PcrV functional activity, while also mediating synergistic protective activity in murine and rabbit lethal acute pneumonia models [24,25]. In addition to its activity in acute infection models, MEDI3902 has shown promising activity against *P. aeruginosa* in animal models where biofilm formation has been described [26,27]. Based on the potency of MEDI3902 in multiple animal models and disease settings, we were interested in testing its activity against *P. aeruginosa* when delivered directly on the ocular surface.

## 2. Materials and Methods 

### 2.1. Pseudomonas Cultures

The lab strain *Pseudomonas aeruginosa* PAO1 was used throughout the experiments described. Prior to inoculation, PAO1 was in grown on lysogeny broth (LB) agar plates and a single colony was selected and grown overnight at 37 °C in LB. Then overnight cultures were diluted to an OD600 of 0.4 which corresponded to an inoculum of roughly 3 × 10^6^ CFUs/eye throughout this study. A higher inoculum of 3 × 10^8^ was used for a high inoculum prevention test, elaborated below.

### 2.2. Animal Studies

#### 2.2.1. Animal Studies and Ethics Statement

All experiments utilizing mice were conducted in strict accordance of the Institutional Animal Care and Use Committee (IACUC) of Dartmouth College, protocol number zega.me.1. For the study, 6–8 week-old C57BL/6 wild-type male and female mice were purchased from The Jackson Laboratory. Twenty mice were used for each study, ten per treatment group.

#### 2.2.2. Corneal Scratch Model

The corneal scratch model was performed as previously described [28,29]. Briefly, mice were anesthetized with 1–2% isoflurane in O_2_ and positioned under an operating microscope. Three scratches were made in the corneal epithelium with a 25-gauge needle with the bevel positioned up. For the non-infectious study designed to determine if the antibodies were toxic to the cornea, no inoculum was given and 5 µL of PBS, MEDI3902 or control IgG (IgG) were given every 2 h (Q2H) for twelve hours a day across seven days and then the corneas were scored. For epithelial damage, a score of 0 or 1 is assigned (0, intact/healed corneal epithelium; 1, non-intact corneal epithelium). For opacity, scores ranging from 0–2 are assigned (0, no opacity; 1, trace opacity with iris unobscured; 2, partial opacity with the iris partially obscured; 3, significant opacity with iris fully obscured; 4, perforated cornea). Additionally, it was noted when other clinical findings were present such as a hypopyon, a hyphema, or perforation of the cornea. Representative images for this scoring are shown in the Appendix A as Appendix A. Those who performed the scratch test and scoring were blinded to the antibodies.

For the infectious studies, an overnight bacterial suspension was used as the inoculum. To confirm the inoculum size, viable bacteria were enumerated by serial dilution and CFUs were counted. For both prevention and treatment studies, mice were given a 5 µL bacterial challenge following corneal scratch. In the prevention study, mice were treated minutes after challenge with 5 µL of antibody (either MEDI3902 or IgG at a concentration of 0.01 mg/µL or 0.05 mg total of antibody), and treatment was given every three hours (Q3H). Mice were then sacrificed the next day after inoculation and treatment. For the prevention study alone, we harvested the corneas of mice and placed them in PBS. Collagenase II was then used to digest them at 37 °C. This was vortexed and the supernatant removed and the rest re-suspended in PBS and serial diluted on LB agar plates and CFUs were counted. In the topical treatment study, mice were not given treatment until 16 h after inoculation in order to let the infection become established. Then, mice were graded for initial presentation of infection and treated with either antibody Q2H for twelve h across three days. At the conclusion of the study, mice were then sacrificed. Additionally, another treatment study was performed where, in place of a topical treatment, mice were given a one-time 200 µL tail vein injection of MEDI3902 or IgG at a final dose of 0.2 mg/kg. Like the topical treatment study, infections were scored at Days 1, 2, and 3 for opacity and epithelial damage. In all studies, eyes were considered uninfected if they had an epithelial and opacity score of 0 at Day 1. At the completion of the experiments, the animals were sacrificed, and treated eyes were then enucleated and fixed in formalin. To understand the general work-flow of each of these experiments, a diagram has been included in Appendix A. In order to facilitate reading, these figures are laid out as well throughout the paper in Figure 1, Figure 2, Figure 3, Figure 4.

### 2.3. Statistical Analysis

For all data sets with the exception of the epithelial scoring, a student’s *t*-test as well as a Mann–Whitney test of significance was performed in Graph-Pad Prism for comparison of groups to determine significance. Only the student *t*-test is shown, however both reported similar significance. For the data involving epithelial scoring, a one-tailed Fisher’s exact test was performed using Excel. Asterisks (*) indicate statistically significant differences (* *p* < 0.05; ** *p* < 0.01) in all figures.

### 2.4. Antibody (MEDI3902) Production

The bispecific antibody (BiS4αPa), commercial name MEDI3902, was provided by MedImmune/AstraZeneca and produced/constructed as previously described [25,30].

## 3. Results

### 3.1. MEDI3902 Shows No in vivo Cytotoxicity to the Surface of the Eye

To determine that MEDI3902 would be safe to use on the cornea of mice with disrupted epithelia, mice were administered MEDI3902, IgG, or PBS following corneal abrasion. After the first day, 50% of the PBS treated, 80% of the MEDI3902 treated, and 60% the IgG treated were healed and all were healed following 7 days (Table 1). None of the groups developed corneal opacities or corneal neovascularization. These data suggest that treatment with MEDI3902 does not cause corneal toxicity or delay corneal epithelial healing.

### 3.2. MEDI3902 Reduced Risk of P. aeruginosa Keratitis when Treatment was Started at the Time of Inoculation

To elucidate the relationship between MEDI3902 and its antimicrobial effects against PAO1 on the ocular surface, we initially treated mice at the moment of inoculation and it was shown that in both a high inoculum (~5 × 10^7^) and a low inoculum (~5 × 10^5^) challenge, MEDI3902 demonstrated preventative capabilities against PAO1 (Figure 1) At Day 1 post-inoculation, mice treated with MEDI3902 had significantly less CFUs (1.83 × 10^2^ in the low inoculum study and 1.56 × 10^4^ in the high) recovered from their corneas than those treated with IgG (5.39 × 10^2^ in the low and 4.63 × 10^4^ in the high) (Figure 2a,b) When the epithelium of the mice were scored, there were four times the number of mice with disrupted epithelial layers in the IgG group, which also proved to be significant (Figure 2c). The infection caused moderate opacities to form in the majority of mice treated with IgG with an average score of 3.0 and 1.3 in the high and low inoculum studies respectively while there was significantly less opacification in mice treated with MEDI3902 with an average score of 2.4 and 0.1 in the high and low studies, respectively (Figure 2d). It is worth noting that while both the high inoculum study and the low inoculum study were reported for CFUs, only the low inoculum study was reported for the corneal scoring. This was due to the fact that the high inoculum was above clinically relevant concentrations [31] and led to such a severe infection and severe damage to the ocular surface early in the course of infection, scoring the cornea with consistency proved to be difficult. The data were shown to be significant but they are now shown here. While neither IgG nor MEDI3902 treated mice developed perforations (this was unlikely given the time-scale of the prevention study), one mouse treated with IgG developed a hyphemia while another developed a hypopyon (Not shown). 

### 3.3. Topical Treatment of MEDI3902 could Speed the Resolution of Established p. Aeruginosa Keratitis

Having noted the ability of MEDI3902 to prevent corneal infection when given at the time of infection, we next investigated whether delivery of the mAb after infection was effective. All mice treated with MEDI3902 showed completely healed epithelium by Day 3 after topical treatment. On Day 1 of treatment, seven of the ten mice treated with MEDI3902 showed no signs of corneal opacity, while three of ten mice treated with IgG had no epithelial defects (Figure 2a). By Day 3 of the infection, all of the mice treated with MEDI3902 had completely re-epithelialized while six of the IgG treated mice had remaining epithelial irregularities. At Day 1, there was also a significant difference between the opacities of the MEDI3902 and IgG treated mice with an average score of 0.3 and 1.7 each. This gap however, decreased over the course of infection as the score for MEDI3902 treated mice rose to 1.0 and IgG mice remained stable around 1.8. (Figure 2b). During the course of infection and treatment, two mice treated with IgG developed ulcers substantial enough to cause severe corneal perforations. No hyphemia or hypopyons were noted during treatment within either cohort. 

### 3.4. A One-Time Intravenous Administration of MEDI3902 Proved Effective at Treating P. aeruginosa Keratitis

While treatment with topical drops demonstrated efficacy in mitigating *p*. aeruginosa keratitis in mice, using an antibody treatment modality allowed us to investigate treating an established PAO1 infection with a one-time intravenous injection. Oral voriconazole has been shown, for example to treat fungal keratitis [32]. These data showed an auspicious outcome for mice treated with MEDI3902. While at Day 1, there was no significant difference between the epithelial integrity of mice treated with either MEDI3902 or IgG, the epithelial cell layer of mice treated with MEDI3902 healed much quicker and by days two and three there was significant difference between the two cohorts. By Day 3 of the treatment, only one of ten IgG treated mice had re-epithelized while seven of ten mice treated with MEDI3902 had completely intact epithelium. (Figure 3a). This trend was consistent in the opacification scores as well. While the IgG and MEDI3902 mice had begun with scores of 2.9 and 2.2 at Day 1, by Day 3 the MEDI3902 cohort decreased to a score of 1.2 and the IgG cohort remained over 2.0. (Figure 3b). Throughout the study two of the mice treated with MEDI3902 had developed a hypopyon but these resolved by Day 3 of treatment. However, six of the mice treated with IgG developed a hyphemia and one had an ulcer which was subsequently perforated.

## 4. Discussion

We have presented our investigations of the use of a bispecific mAb targeting *P. aeruginosa* Psl and PcrV in the treatment of *P. aeruginosa* keratitis. Our data indicate that immunotherapy is well tolerated on the ocular surface and can inhibit the development of infection when given concurrently at the time of corneal inoculation and can speed resolution of *P. aeruginosa* keratitis compared to IgG when administered either topically or IV after an established infection. It is important to note that treatment of mice with eye drops is inherently challenging compared to human treatment. Patient instructions to avoid blinking and perform digital compression of the nasolacrimal system after instillation of drops are all strategies which have been effective in improving corneal absorption of topical medications. Future use of mAb-based approaches for this indication will require optimization of factors that promote corneal penetration and retention on the ocular surface [33]. Previously developed strategies for other topical medications include the use of surfactants to reduce the surface tension of drops on the ocular surface. Viscus compounds such as methylcellulose have often been included in eye drops to promote longer dwell time on the ocular surface and thus increase corneal absorption of medications [34]. Such formulations may enhance the efficacy of topical mAbs on the ocular surface.

Intravenous use of antimicrobial antibodies has been proven effective for some infections such as Bezlotoxumab [35] in the treatment of *Clostridium difficile* toxin B and palivizumab [36] for respiratory syncytial virus and, as previously mentioned, MEDI3902 has been shown effective in *Pseudomonas* keratitis prophylaxis [3]. Since they are unconventional antimicrobials, microbes resistant to conventional antibiotics are less likely to have pre-existing resistance mechanisms to mAbs. Unlike antibiotics which are typically cleared from the blood in hours to days, antibodies have a half-life which is typically measured in weeks to months [37]. For these reasons, we would envision the possible use of MEDI3902 or similar mAbs in patients with a chronic *P. aeruginosa* keratitis who are unable to comply with use of eye drops. This may be desirable compared to prolonged inpatient admission or nurse administration of eye drops. Patients with *P. aeruginosa* strains which are resistant to conventional anti-*Pseudomonal* medications or patients and who are allergic to these medications would also be candidates for treatment with an anti-*Pseudomonal* mAb.

Important challenges in the clinical use of MEDI3902 for keratitis exists. Unlike a broad-spectrum antibiotic, MEDI3902 is a bispecific mAb against *P. aeruginosa* which targets the exopolysaccharide Psl as well as PcrV, a component of the Type III secretion system. We previously reported that 99% of the *Pseudomonas* isolates from the Steroids for Corneal Ulcer Trial (SCUT) produced Psl indicating that the overwhelming majority of clinical isolates of *Pseudomonas* would be susceptible to mAbs targeting Psl [15]. That said, non Psl producing strains are known to exist and patients infected with these strains would not respond to MEDI3902 unless the anti-PcrV mAb is sufficient for efficacy. However, previous work suggests that there is some efficacy from mAbs which only target PcrV [38] and a global surveillance study of *P. aeruginosa* strains determined that only 1 out of over 900 isolates lacked both targets [26]. Theoretically, this mAb could provide >99.9% coverage based on this previous work. Still, MEDI3902 would not have an application as an empiric treatment for an ulcer in which *P. aeruginosa* had not been established as causative agent. Its role in keratitis therapy currently would be focused on drug resistant and/or chronic *P. aeruginosa* infections. While this study did not investigate the efficacy in polymicrobial keratitis [39,40], in which *P. aeruginosa* is one of several agents present. Future investigations can be done to determine its efficacy when given in combination with various broad-spectrum antibiotics which has been [41]. For example, tobramycin is often the standard for *P. aeruginosa* infections tobramycin’s efficacy in the treatment of keratitis is limited [42], a previous study investigated the efficacy of tobramycin and MEDI3902 in combination [3]. Further investigation of possible combination treatments is worthy of consideration. Critical to the discussion of MEDI3902’s efficacy is its comparison to moxifloxacin, which has long been the gold standard broad-spectrum antibiotic for the treatment of keratitis [43]. Proven effective in both standard cases [44] and atypical cases [45] moxifloxacin’s prevalence and reliability as the current standard are irrefutable. Given the increasing prevalence of resistance to fluoroquinolones [46], having a pathogen specific alternative is a help and not a hurt; however, it is still critical to compare the efficacy of MEDI3902 to moxifloxacin. 

## 5. Conclusions

In brief, the bispecific antibody MEDI3902 targeting PcrV and Psl is non-toxic to the ocular surface and effective at preventing and treating *Pseudomonas aeruginosa* keratitis in mice when given both topically and intravenously. This was performed using a corneal scratch model. Given the increase in multidrug resistant strains of bacteria concurrent with a scarcity of new antimicrobial agents to counteract this rise as mentioned, the promise of a novel treatment modality for microbial keratitis is auspicious. In addition, given the near ubiquity of both Psl and PcrV within the global population of *Pseudomonas aeruginosa*, the specificity of this antibody in *Pseudomonas* keratitis is in no way a limitation of its efficacy but rather strengthen it. The possibility of a one-time intravenous administration of the drug as opposed to the need to regularly instill drops is noteworthy, especially in patients with difficulty instilling drops. Optimization regarding drop formulation is necessary before use on a human eye. Future studies would be designed to compare the monoclonal’s efficacy compared to current standards of treatment for keratitis (EG moxifloxacin) as well as its effectiveness in a polymicrobial infection, perhaps as part of a combination treatment.

## Figures and Tables

**Figure 1 vaccines-08-00638-f001:**
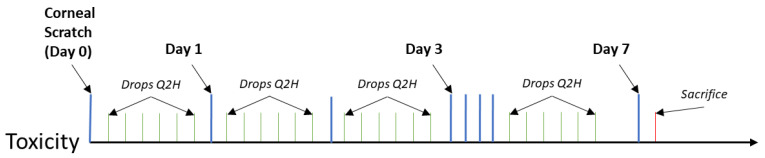
Timescale for cytotoxity study: Mice were given an abrasion on Day 0. Drops of IgG or MEDI3902 were administered Q2H (every two hours) for seven days. Mice were scored and sacrificed on Day 7 after abrasion.

**Figure 2 vaccines-08-00638-f002:**
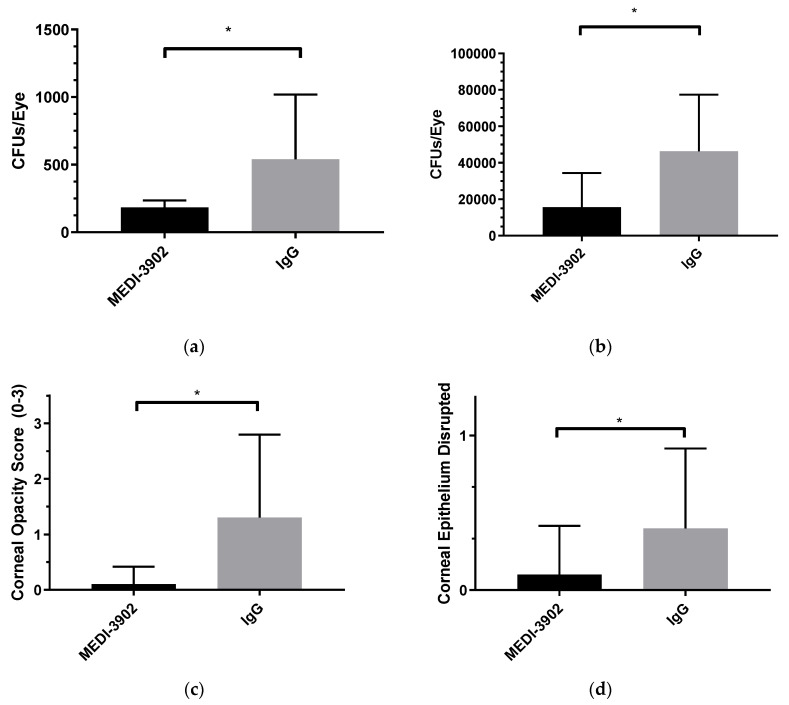
MEDI3902 can prevent an infection of *Pseudomonas aeruginosa *(PAO1) keratitis from occurring when given at the time of inoculation: Mice were challenged with both a low inoculum (~5 × 10^5^) (**a**,**c**,**d**) and a high inoculum (~5 × 10^7^) (**b**) of PAO1 following corneal scratch and then subsequently treated with IgG (*n* = 10) or MEDI3902 (*n* = 10). Mice treated with MEDI3902 had fewer CFUs retrieved one day status post inoculation (**a**,**b**), decreased corneal opacity (**c**), and fewer disrupted epithelia (**d**) versus their IgG counterparts. (Mean ± SEM) MEDI3902 treated low = 183.6 ± 16.6; IgG treated low = 539.3 ± 151.4; MEDI3902 treated high = 15,620 ± 5918; IgG treated high = 46,330 ± 9818; Opacity score: MEDI3902 low 0.1 ± 0.1; IgG low 1.3 ± 0.5. Epithelial score: MEDI3902 low 0.1 ± 0.1; IgG low 0.4 ± 0.2. Epithelial and opacity scores for high not shown. (**a**) Low inoculum; (**b**) high inoculum; (**c**) low inoculum; (**d**) low inoculum; (**e**) timescale.

**Figure 3 vaccines-08-00638-f003:**
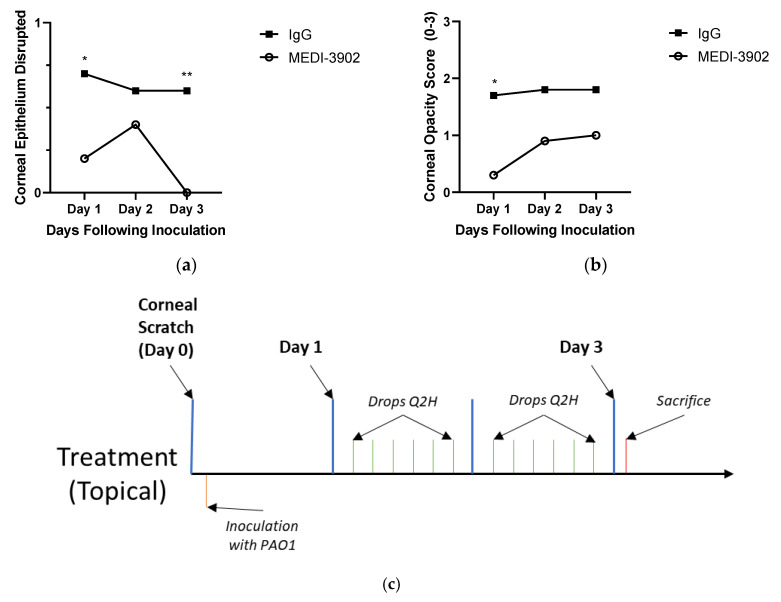
MEDI3902 can treat an established infection of PAO1 when given topically: Mice treated with MEDI3902 (*n* = 10) had fully intact epithelia by day three status post treatment as opposed to those treated with IgG (*n* = 10) (**a**). Similarly, mice treated with MEDI3902 had an initial decrease in opacity one day status post treatment (**b**); however, this was no longer significant by Day 3. (Mean ± SEM) Opacity scores: MEDI3902 treated Day 1 = 0.3 ± 0.153; IgG treated Day 1 = 1.7 ± 0.473; MEDI3902 treated Day 2 = 0.9 ± 0.407; IgG treated Day 2 = 1.8 ± 0.490; MEDI3902 treated Day 3 1.0 ± 0.422; IgG treated Day 3 1.8 ± 0.490. Epithelial scores: MEDI3902 treated Day 1 = 0.2 ± 0.133; IgG treated Day 1 = 0.7 ± 0.153; MEDI3902 treated Day 2 = 0.4 ± 0.163; IgG treated Day 2 = 0.6 ± 0.163; MEDI3902 treated Day 3 0.0 ± 0.0; IgG treated Day 3 = 0.6 ± 0.163. (**c**) Timescale.

**Figure 4 vaccines-08-00638-f004:**
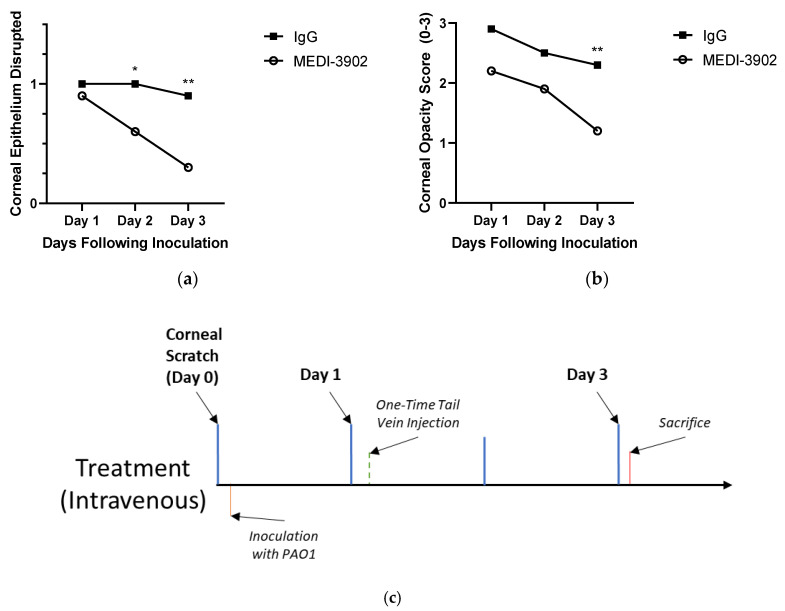
MEDI3902 can treat PAO1 keratitis when given as a one-time, intravenous injection through the tail vein of mice: Mice with active *Pseudomonas* keratitis treated with a one-time, intravenous injection of MEDI3902 (*n* = 10) saw a decrease in the number of mice with disrupted epithelia (**a**) and a decrease in the overall corneal opacity (**b**) throughout three days observation following treatment compared to the the IgG treated mice (*n* = 10). (Mean ± SEM) Opacity scores: MEDI3902 treated Day 1 = 2.2 ± 0.327; IgG treated Day 1 = 2.9 ± 0.100; MEDI3902 treated Day 2 = 1.9 ± 0.314; IgG treated Day 2 = 2.5 ± 0.167; MEDI3902 treated Day 3 1.2 ± 0.200; IgG treated Day 3 2.3 ± 0.213. Epithelial scores: MEDI3902 treated Day 1 = 0.9 ± 0.1; IgG treated Day 1 = 1.0 ± 0.0; MEDI3902 treated Day 2 = 0.6 ± 0.163; IgG treated Day 2 = 1.0 ± 0.0; MEDI3902 treated Day 3 = 0.3 ± 0.153; IgG treated Day 3 = 0.9 ± 0.100. (**c**) Timescale.

**Table 1 vaccines-08-00638-t001:** MEDI3902 is not cytotoxic to the cornea and does not impede restoration of the corneal epithelial layer: One cornea for each mouse was disrupted using the corneal scratch method and then mice were given drops, either PBS, IgG, or MEDI3902, for one week. After one week, all mice had intact epithelium, demonstrating that the drops were not toxic to the corneal surface at experimental concentrations.

Percentage of Mice with Intact Epithelium			
	PBS	MEDI-3902	IgG
Day 1	50%	80%	60%
Day 7	100%	100%	100%

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
