# Peer review of "Multifunctional Monoclonal Antibody Targeting *Pseudomonas aeruginosa* Keratitis in Mice"

_vaccines, 2020, doi:10.3390/vaccines8040638_

Round 1

Reviewer 1 Report

In this manuscript the safety and efficacy of MEDI3902, a bispecific mAb against P. aeruginosa, which targets the exopolysaccharide Psl as well as PcrV, a component of the Type III secretion system on ocular infections. The authors demonstrated that treatment with the antibody is well tolerated on the ocular surface. MEDI3902 antibody treatment can inhibit the development of infection when given concurrently at the time of corneal inoculation and can speed resolution of P. aeruginosa keratitis compared to IgG when administered either topically or IV after an established infection.

There is a real need for novel broad-spectrum antimicrobials in a variety of bacterial infections, including for the treatment of ocular surface infections. This need raises from the limited number of antimicrobials approved for use on the ocular surface and the increase, in recent years, in multidrug resistant strains of bacteria, not responding to known anti bacterial agents. Thus, the research topic is important. However, there some comments regarding this manuscript that should be addressed.

MEDI3902 antibody; the antibody tested in this study, is hardly described in the manuscript; briefly mentioned in the abstract and the discussion:

The source (company producing this antibody) should be stated in the Materials and Methods sections.

In the Introduction section; the antibody should be mentioned and described in details and a deep survey of the literature for its use; mainly in in vivo mice models for infections, should be cited.

Mainly; the dose of the antibody given in all experiments should be stated clearly; 5uL of drops or 200uL for injections are not informative enough; should be given in concentration units (its a protein that its concentration can be measured!)

The stain of the mice used for this study is not disclosed; please indicate the exact mouse stain used in the material and method section. In this regard, number of mice used for the toxicity study; please indicate this important information.

Significance of results: although the authors stated that the results are significant; the diversity of both control and treated mice is very high as demonstrated by the ±SEM value in Figure 1. The authors should go back to the raw data and test the significance utilizing an additional test; one-way ANOVA maybe???

In this regard, Figures 2&3 should show variation (±SEM or ±SE) for each point of the graph.

What indicates ** in the legends of Figures 2&3?; please add information.

Figure 1: corneal opacity (c), disrupted epithelia; were both measured and examined in mice receiving low inoculum of bacteria; authors should add the results of the treated mice versus control mice getting a high inoculum of bacteria; if there is no improvement in those measurements, it should be stated and explained; as it question the real efficacy of the treatment with the antibody.

Figure 2: Each point in both graphs should be given with its statistics; mean ± SEM (or mean ± SD).

Supplementary Figure 2: The Figure is indeed informative. However, as the treatment schedule for each experiment is different and it’s hard o follow while reading the manuscript; it is highly recommended to split this Figure and add the relevant schematic schedule at the beginning of each Figure in the result section.

Abstract: It is stated: “We report the first use of an immunotherapeutic multifunctional agent against, targeting Psl and type 3 secretion, on the ocular surface as an antimicrobial agent. One should really wonder regarding this strong statement as one of the co-authors of the present manuscript is a co-author of a paper already published in 2019; demonstrating, among other results, that “P. aeruginosa Biofilm Formation Can Be Blocked In Vivo Using a Bispecific Antibody Targeting Both T3SS and Psl” using the same MEDI3902 antibody and the same ocular Keratitis Model; see Cell Host Microbe,

2019, 25(4):526-536. The authors should omit the phrase “for the first time” and add this reference to the manuscript.

Author Response

There is a real need for novel broad-spectrum antimicrobials in a variety of bacterial infections, including for the treatment of ocular surface infections. This need raises from the limited number of antimicrobials approved for use on the ocular surface and the increase, in recent years, in multidrug resistant strains of bacteria, not responding to known anti-bacterial agents. Thus, the research topic is important. However, there some comments regarding this manuscript that should be addressed.

MEDI3902 antibody; the antibody tested in this study, is hardly described in the manuscript; briefly mentioned in the abstract and the discussion:

The authors deliberated on whether to include a more extensive section on this very idea prior to this suggestion in an early draft of the paper and initially decided not to due to extensive literature put forth by one of the authors already available. Per the reviewer’s suggestion, we decided to incorporate a significant literature review in the end of the introduction of the paper on MEDI3902.

The source (company producing this antibody) should be stated in the Materials and Methods sections.

We included the company (MedImmune/AstraZeneca) in the M&M section as requested by the reviewer.

In the Introduction section; the antibody should be mentioned and described in details and a deep survey of the literature for its use; mainly in in vivo mice models for infections, should be cited.

My co-author AD, who provided the antibody, added a literature review of MEDI3902 at the conclusion of the introduction.

Mainly; the dose of the antibody given in all experiments should be stated clearly; 5uL of drops or 200uL for injections are not informative enough; should be given in concentration units (it’s a protein that its concentration can be measured!)

 The authors do appreciate the reviewer’s suggestion here as this was missed during our initial edits. We added the final concentrations of antibody, 0.2 mg/kg intravenously and 0.01mg/uL for topical administration (0.05mg total) to the Material and Methods.

The stain of the mice used for this study is not disclosed; please indicate the exact mouse stain used in the material and method section. In this regard, number of mice used for the toxicity study; please indicate this important information.

Per the reviewer’s suggestion, we have added the strain type and relevant information to section 2.2.1.

Significance of results: although the authors stated that the results are significant; the diversity of both control and treated mice is very high as demonstrated by the ±SEM value in Figure 1. The authors should go back to the raw data and test the significance utilizing an additional test; one-way ANOVA maybe???

Per the reviewer’s comments we did do additional statistics (specifically a Mann-Whitney test. A one-way ANOVA needs 3 or more groups and we only had two: MEDI3902 and IgG). These statistics all corroborate with our results and in some cases report significance (or greater significance, IE lower p values) where the student’s t-test did not. We shared this in the statistical methods section but did not report the statistics since the student’s t-test was the most appropriate test of significance here. I reported the Mean±SEM for each data point as requested. Indicated in the statistical methods that * = p<0.05 and ** = p<0.01 per reviewer.

In this regard, Figures 2&3 should show variation (±SEM or ±SE) for each point of the graph.

± SEM has been added to the graphs on figures 2 and 3, per the reviewer's other suggestion. The version of the paper which was uploaded didn't seem to have them, so I have attached the properly revised version here!

What indicates ** in the legends of Figures 2&3?; please add information.

* and ** (p<0.05 and p<0.01, respectively) was stated in the M&M section under the statistical analysis.

Figure 1: corneal opacity (c), disrupted epithelia; were both measured and examined in mice receiving low inoculum of bacteria; authors should add the results of the treated mice versus control mice getting a high inoculum of bacteria; if there is no improvement in those measurements, it should be stated and explained; as it question the real efficacy of the treatment with the antibody.

The reviewer is correct that the CFUs for both the low and high inoculum were reported; however, the corneal scores for the high inoculum were not reported. This was because the high inoculum dose challenge was far above clinical relevance and the corneas for both IgG and MEDI3902 treated mice were severely damaged early in the infection course. While this was shown to be significant, we felt that the scoring was difficult and at times inconsistent due to the severity of damage to the ocular surface.  Per the reviewer, this was exposited in the text of the manuscript.

Figure 2: Each point in both graphs should be given with its statistics; mean ± SEM (or mean ± SD).

The mean ± SEM was provided for every figure for the paper, per the reviewers request.

Supplementary Figure 2: The Figure is indeed informative. However, as the treatment schedule for each experiment is different and it’s hard to follow while reading the manuscript; it is highly recommended to split this Figure and add the relevant schematic schedule at the beginning of each Figure in the result section.

We incorporated the timescales for each experiment in along with the corresponding experimental discussion. This indeed made the process far easier to follow, as the reviewer said. We elected to leave the supplemental figure in the paper as it gives an overall view of the differences for each process.

Abstract: It is stated: “We report the first use of an immunotherapeutic multifunctional agent against, targeting Psl and type 3 secretion, on the ocular surface as an antimicrobial agent. One should really wonder regarding this strong statement as one of the co-authors of the present manuscript is a co-author of a paper already published in 2019; demonstrating, among other results, that “P. aeruginosa Biofilm Formation Can Be Blocked In Vivo Using a Bispecific Antibody Targeting Both T3SS and Psl” using the same MEDI3902 antibody and the same ocular Keratitis Model; see Cell Host Microbe,

This was a very helpful study to read through and incorporating engagement with it into the paper was very helpful and proved to improve the quality of the paper. Per the reviewer, we altered the language in the abstract to provide a clearer picture of what our particular study offers to the field. We also engaged with this study a few more times throughout the paper as well. The key differences between our study and this one addressed: they showed that MEDI3902 when given prophylactically was effective at preventing biofilm formation whilst ours was given after an infection was established, they used MEDI3902 solely as an intravital treatment whilst we used it intravitally and topically, they showed that MEDI3902 when given in combination with tobramycin was effective as a treatment which was helpful given the further interest in combination therapies.

2019, 25(4):526-536. The authors should omit the phrase “for the first time” and add this reference to the manuscript.

We updated the language in the abstract to more accurately describe the contribution of this paper. While not the first use of MEDI3902 intravitally to prevent PA keratitis, we are the first to use it topically and intravitally to treat (not prevent).

We are very appreciative of this reviewer’s thorough reading of our paper. Overall, the authors feel that the revisions made as a result improve the quality of the paper as a whole. We hope that these extensive revisions are sufficient for consideration.

Reviewer 2 Report

The paper is well written and I think this is an important study in the space of keratitis treatment.

I do not have any major issues with the paper in its current form. 

My only concern with this article is:

This is written from the point of view of a 'first-of-its-kind' treatment protocol, however, the authors themselves mention that moxifloxacin is a drug that already is the current commercial standard. 

To this end, it is really important to have a side-by-side comparison with either moxifloxacin or any other competent drug for the same, to throw some light about the efficiency of the procedure. Even if the real numbers show that this strategy work less optimally than commercial drug concoctions, it would still remain an important contribution as an alternate therapeutic approach.

If the authors find it difficult to re-do experiments with another commercial drug/ Ab, they should at least provide a discussion on the same in reference to articles such as the following:

1)https://iovs.arvojournals.org/article.aspx?articleid=2383645

2)https://www.healio.com/news/ophthalmology/20160112/persistent-infectious-keratitis-in-a-contact-lens-wearer

3)https://academic.oup.com/cid/article/54/10/1381/350890

4)https://citeseerx.ist.psu.edu/viewdoc/download?doi=10.1.1.446.1452&rep=rep1&type=pdf

etc.

Author Response

My only concern with this article is:

This is written from the point of view of a 'first-of-its-kind' treatment protocol, however, the authors themselves mention that moxifloxacin is a drug that already is the current commercial standard. 

To this end, it is really important to have a side-by-side comparison with either moxifloxacin or any other competent drug for the same, to throw some light about the efficiency of the procedure. Even if the real numbers show that this strategy work less optimally than commercial drug concoctions, it would still remain an important contribution as an alternate therapeutic approach.

If the authors find it difficult to re-do experiments with another commercial drug/ Ab, they should at least provide a discussion on the same in reference to articles such as the following:

1)https://iovs.arvojournals.org/article.aspx?articleid=2383645

2)https://www.healio.com/news/ophthalmology/20160112/persistent-infectious-keratitis-in-a-contact-lens-wearer

3)https://academic.oup.com/cid/article/54/10/1381/350890

4)https://citeseerx.ist.psu.edu/viewdoc/download?doi=10.1.1.446.1452&rep=rep1&type=pdf

Reviewer 2 had one primary critique of the paper, its lack of engagement with moxifloxacin as the current commercial standard for keratitis treatment. The authors agree that this is lacking and so per the reviewer’s suggestion, the discussion was modified to include substantial interaction with the need for further studies comparing the efficacy of MEDI3902 to moxifloxacin as the current gold standard in the treatment of bacterial keratitis. The authors agree with the reviewer that it would have been ideal to conduct further studies comparing these two. We were limited however by the first author’s change in career and role in the clinic/lab. Overall, this revision acts to improve the overall quality of the paper and therefore the authors are grateful to the reviewer for their suggestion.

“Critical to the discussion of MEDI3902’s efficacy is it’s comparison to moxifloxacin, which has long been the gold standard broad-spectrum antibiotic for the treatment of keratitis[41]. Proven effective in standard cases[42], atypical cases[43], and persistent cases[44], moxifloxacin’s prevalence and reliability as the current standard are irrefutable. Given the increasing prevalence of resistance to fluoroquinolones[45], having a pathogen specific alternative is a help and not a hurt; however, it is still critical to compare the efficacy of MEDI3902 to moxifloxacin.”

Round 2

Reviewer 1 Report

The authors addressed the comments and corrected accordingly the manuscript. 

Reviewer 2 Report

It can be accepted now! Reads better ...